# Supervector Extraction for Encoding Speaker and Phrase Information with Neural Networks for Text-Dependent Speaker Verification †

**Victoria Mingote \*** , **Antonio Miguel \***, **Alfonso Ortega \*** **and Eduardo Lleida \***

ViVoLab, Aragón Institute for Engineering Research (I3A), University of Zaragoza, 50018 Zaragoza, Spain
* Correspondence: vmingote@unizar.es (V.M.); amiguel@unizar.es (A.M.); ortega@unizar.es (A.O.);
  lleida@unizar.es (E.L.)
† This paper is an extended version of our paper published in the conference IberSPEECH2018.

**Abstract:** In this paper, we propose a new differentiable neural network with an alignment mechanism for text-dependent speaker verification. Unlike previous works, we do not extract the embedding of an utterance from the global average pooling of the temporal dimension. Our system replaces this reduction mechanism by a phonetic phrase alignment model to keep the temporal structure of each phrase since the phonetic information is relevant in the verification task. Moreover, we can apply a convolutional neural network as front-end, and, thanks to the alignment process being differentiable, we can train the network to produce a supervector for each utterance that will be discriminative to the speaker and the phrase simultaneously. This choice has the advantage that the supervector encodes the phrase and speaker information providing good performance in text-dependent speaker verification tasks. The verification process is performed using a basic similarity metric. The new model using alignment to produce supervectors was evaluated on the RSR2015-Part I database, providing competitive results compared to similar size networks that make use of the global average pooling to extract embeddings. Furthermore, we also evaluated this proposal on the RSR2015-Part II. To our knowledge, this system achieves the best published results obtained on this second part.

**Keywords:** text-dependent speaker verification; HMM alignment; deep neural networks; supervectors

## 1. Introduction

Recently, techniques based on discriminative deep neural networks (DNN) have achieved substantial success in many speaker verification tasks. These techniques follow the philosophy of the state-of-the-art face verification systems [1,2] where embeddings are usually extracted by reduction mechanisms and the decision process is based on a similarity metric [3]. Unfortunately, in text-dependent tasks, this approach does not work efficiently since the pronounced phrase is part of the identity information [4,5]. A possible cause of the inaccuracy in text-dependent tasks could be derived from using the temporal average as a representation of the whole utterance as we show in the experimental section. To solve this problem, this paper presents a new architecture which combines a deep neural network with a phonetic phrase alignment method used as a new internal layer to maintain the temporal structure of the utterance. As we will show, it is a more natural solution for the text-dependent speaker verification since the speaker and phrase information can be encoded in the supervector thanks to the neural network and the specific states of the supervector. A supervector is a concatenation of smaller-dimensional vectors from each specific state into a higher-dimensional vector.

In the context of text-independent speaker verification tasks, the baseline system based on i-vector extraction and Probabilistic Linear Discriminant Analysis (PLDA) [6,7] is still among the best results of the state-of-the-art. An i-vector is a representation of an utterance in a low-dimensional subspace called the total variability subspace, and the PLDA model produces the verification scores. However, as we previously mentioned, many improvements on this baseline system have been achieved in recent years by progressively substituting components of the systems by DNNs, thanks to their greater expressiveness and the availability of large databases. Examples of this are the use of DNN bottleneck representations as features replacing or combined with spectral parametrization [8], training DNN acoustic models to use their outputs as posteriors for alignment instead of Gaussian Mixture Model (GMM) in i-vector extractors [9], or replacing PLDA by a DNN [10]. Other proposals similar to face verification architectures have been more ambitious and have trained a discriminative DNN for multiclass classification. Once these systems are trained, the embeddings, which are fixed-length utterance level representations, are extracted by reduction mechanisms from the DNN [11–13]. In this context, they are called x-vectors—for example, taking the average of an intermediate layer usually named the bottleneck layer. After that embedding extraction, the verification score is obtained by a similarity metric such as cosine similarity [11] or PLDA.

The application of DNNs and the same techniques as in text-independent models for text-dependent speaker verification tasks has produced mixed results. On the one hand, specific modifications of the traditional techniques have been shown to be successful for text-dependent tasks such as i-vector+PLDA/Support Vector Machines (SVM) [14–16], DNNs bottleneck as features for i-vector extractors [17] or posterior probabilities for i-vector extractors [17,18]. On the other hand, speaker embeddings obtained directly from a DNN have provided good results in tasks with large amounts of data and a single phrase [19], but they have not been as effective in tasks with more than one pass phrase and smaller database sizes [4,5]. The lack of data in this last scenario may lead to problems with deep architectures due to overfitting of models.

Another reason that we explore in the paper for the lack of effectiveness of these techniques in general text-dependent tasks is that the phonetic content of the uttered phrase is relevant for the identification. State-of-art text-independent approaches to obtain speaker embeddings from an utterance usually reduce temporal information by calculating the average across frames of the internal representations of the network. This approach may neglect the order of the phonetic information because, in the same phrase, the beginning of the sentence may be totally different from what is said at the end. An example of this is the case when the system asks the speaker to utter digits in some random order. In that case, a mean vector would fail to capture the combination of phrase and speaker. Therefore, one of the objectives of the paper is to show that it is important to keep this phrase information for the identification process, not just the information of who is speaking.

In previous works, we have developed systems that need to store a model per user which were adapted from a universal background model. These systems were developed using two approaches: a statistical model based on Factor Analysis (FA) [20], and a model based on Neural Networks [21]. In both systems, the evaluation of the trial was a likelihood ratio. One of the drawbacks of this approach is the need to store a large amount of data per user and the speed of evaluation of trials since likelihood expressions were dependent on the frame length. To simplify this evaluation process, here we employ a cosine similarity as a metric. This metric is simple and allows us to evaluate the trials using a matrix structure, which reduces the time needed to make the whole evaluation.

In this paper, we propose a new approach that includes alignment as a key component of the mechanism to obtain the vector representation from a deep neural network. Unlike previous works, we substitute the average of the internal representations across time that is used in other neural network architectures [4,5] by an alignment mechanism to keep the temporal structure of each utterance. We show how the alignment can be applied in combination with a DNN acting as a front-end to create a supervector for each utterance. As we will show, the application of both sources

of information in the process of defining the supervector provides better results in the experiments performed on RSR2015 [22] compared to previous approaches.

This paper is organized as follows. In Section 2, we present our system and especially the alignment strategy developed. Section 3 presents the experimental data. Section 4 explains the results achieved. Conclusions are presented in Section 5.

## 2. Deep Neural Network Based on Alignment

Due to the poor performance of the results achieved in previous works for this task with only DNNs and a basic similarity metric, we have combined the neural network with a differentiable alignment mechanism due to the importance of the phrases and their temporal structure in these kinds of tasks. Since the same person does not always pronounce one phrase at the same speed or in the same way due to differences in the phonetic information, it is usual that there exists an articulation and pronunciation mismatch between two compared speech utterances even from the same person. Thus, we have developed this approach to give robustness to the systems against these sources of variability.

### 2.1. System Description

Figure 1 shows the overall architecture of our system which consists of a front-end part, where the acoustic features are obtained from the input signal, and a neural network is applied over them. After that, the global average pooling usually employed to obtain the vector embedding before the back-end is substituted by the alignment process to finally create a supervector per audio file. This supervector can be seen as a mapping between an utterance and the state components of the alignment, which allows encoding the phrase information. These two parts are combined, and a flatten layer is used to link with the last layer to train the system for multi-class classification. For the verification process, once our system is trained, one supervector is extracted from the flatten layer for each enroll and test utterance, and then a cosine metric is applied over them to produce the verification scores.

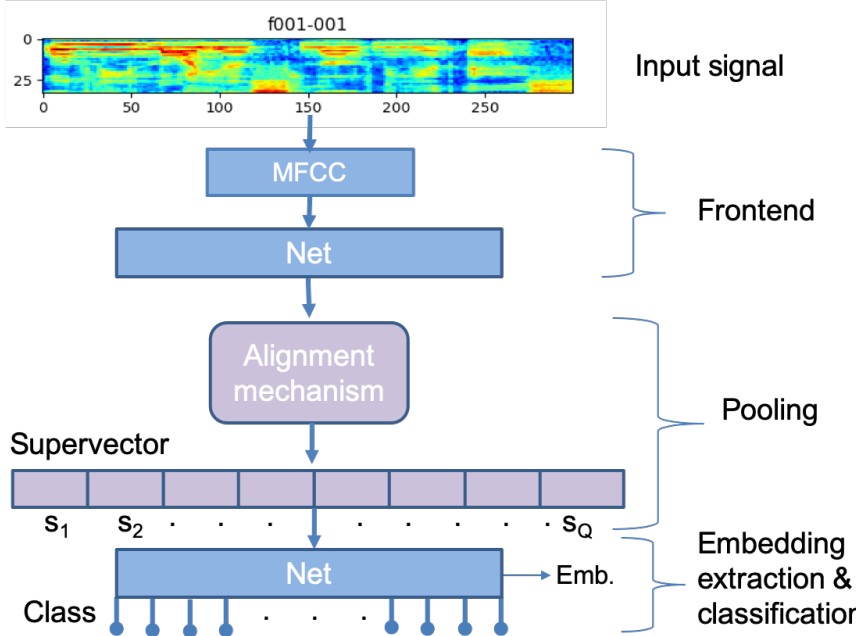

**Figure 1.** Differentiable neural network alignment mechanism based on alignment models. The supervector is composed of Q vectors $s_Q$ for each state.

## 2.2. Front-End Network Architecture

For deep speaker verification, some simple architectures with only dense layers [4] have been proposed. However, lately, employed deep neural networks such as Residual Convolutional Neural Networks (CNN) [5], but, in text-dependent tasks, it has not achieved the same good results as previous simple approaches.

In our network, we propose a straightforward architecture with only a few layers that include the use of one-dimensional convolution (1D convolution) layers instead of dense layers or 2D convolution layers as in other works. Our proposal is to operate in the temporal dimension to add context information to the process and at the same time the channels are combined at each layer. The context information that is added depends on the size of the kernel $k$ used in the convolution layer.

To use this type of layer, it is convenient that the input signals have the same size. For this reason, we apply a transformation to interpolate the input signals to have all of them with the same dimensions $(D^0 \times T)$ where $D^0$ is the acoustic feature dimension of the Mel-Frequency Cepstral Coefficients (MFCC) with their first and second derivatives, and $T$ is the temporal dimension.

The operation of the 1D convolution layers is depicted in Figure 2, and the layer input and its context, the previous frames and the subsequent frames are multiplied frame by frame with the corresponding weights. The results of this operation for each frame are linearly combined to create the output signal. Using this kind of layer in the network architecture, the acoustic feature dimension of the input signal changes after each convolution layer. As we can see in Figure 2, each convolution layer will have feature vectors of dimension $D^{(l)}$ as input and $D^{(l+1)}$ as output. The resulting dimension of the processed signal through the network will be $D = (D^{(0)}, D^{(1)}, ..., D^{(l)}, ..., D^{(L)})$, where $L$ is the total number of front-end layers. The temporal dimension $T$ is kept constant in these layers since we define a zero padding for the convolution.

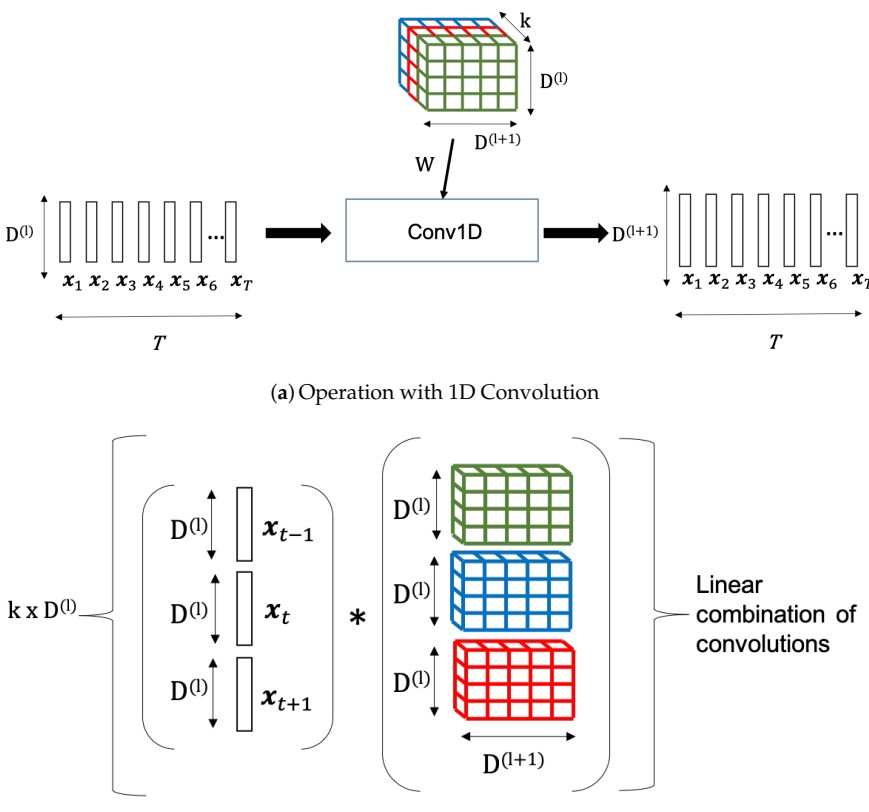

(**a**) Operation with 1D Convolution

(**b**) Example of the convolution operation

**Figure 2.** Operation with 1D Convolution layers, (**a**) general pipeline of this operation; (**b**) example of how k context frames from input are multiplied by the weight matrix W and the output is equivalent to a linear combination of convolutions.

*2.3. Alignment Mechanism*

In this work, we select a Hidden Markov Model (HMM) as the alignment technique in all the experiments. In text-dependent tasks, we know the phrase transcription that allows us to construct a specific left-to-right HMM with no skip states for each phrase of the data and obtain a Viterbi alignment per utterance.

One reason to employ a phrase HMM alignment was due to its simplicity for training independent HMMs for different phrases used to develop our experiments without the need of phonetic information for training. Another reason was that, using the decoded sequence provided by the Viterbi algorithm in a left-to-right architecture, it is ensured that each state of the HMM corresponds to at least one frame of the utterance, so no state is empty.

The process followed to add this alignment to our system is detailed below. Once the models for the alignment are trained, a sequence of decoded states $\bar{q} = (q_1, ..., q_t, ..., q_T)$ where $q_t$ indicates the decoded state at time t with $q_t \in \{1, ..., Q\}$ is obtained. As we can see in Figure 3, we encode this information using an alignment matrix $A \in \mathbb{R}^{T \times Q}$ where $a_{tq_t} = 1$ if frame $t$ belongs to state $q_t$ and 0 otherwise. Since only one state is active at the same time, $\sum_q a_{tq} = 1$. Thanks to this matrix encoding, we can include the alignment information in the gradient estimation increasingly.

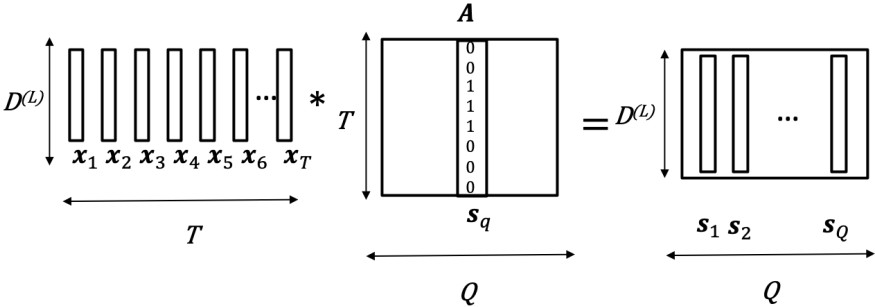

**Figure 3.** Process of alignment, the input signal $x$ is multiplied by an alignment matrix A to produce a matrix with vectors $s_Q$ which are then concatenated to obtain the supervector.

For example, if we train an HMM with four states, we obtain a vector $\bar{q}$ and apply the previous transformation, the resultant matrix $A$ would be:

$$\bar{q} = [1, 1, 1, 2, 2, 3, 3, 4] \rightarrow A = \begin{pmatrix} 1 & 0 & 0 & 0 \\ 1 & 0 & 0 & 0 \\ 1 & 0 & 0 & 0 \\ 0 & 1 & 0 & 0 \\ 0 & 1 & 0 & 0 \\ 0 & 0 & 1 & 0 \\ 0 & 0 & 1 & 0 \\ 0 & 0 & 0 & 1 \end{pmatrix}. \tag{1}$$

The input employed to this alignment layer is the output of the $L$ convolutional layer. The alignment as a matrix multiplication can be expressed as a function of the input signal, $x_{dt}^L$, with dimensions $(D^{(L)} \times T)$ and the alignment matrix of each utterance $A$ with dimensions $(T \times Q)$:

$$x_{dq}^{(L+1)} = \frac{\sum_t x_{dt}^{(L)} \cdot a_{tq}}{\sum_t a_{tq}}, \tag{2}$$

where $x_{dq}^{(L+1)}$ is the supervector of the layer $(L + 1)$ with dimensions $(D^{(L)} \times Q)$, where there are $Q$ state vectors of dimension $D^{(L)}$ and we normalize by the number of times state $q$ is active.

## 3. Experimental Setup

### 3.1. Data

In all the experiments in this paper, we used the RSR2015 text-dependent speaker verification dataset [22]. This dataset consists of recordings from 157 male and 143 female, and nine different sessions for each speaker. This data are divided into three speaker subsets: background (bkg), development (dev) and evaluation (eval). Furthermore, it is also divided into three parts based on different lexical constraints. We develop our experiments in Part I and Part II of this data set. Part I is based on 30 phonetically balanced pass-phrases. Part II contains 30 short commands to control. The average length of the utterances in the second part is half the length of the utterances in Part I. With both parts, we employ the bkg and dev data (194 speakers, 94 female/100 male) for training. The evaluation part is used for enrollment and trial evaluation.

This dataset has three evaluation conditions, but, in this work, we have only evaluated the Impostor-Correct case that is the most challenging condition where the non-target speakers pronounce the same phrase as the target speakers.

### 3.2. Experimental Description

In our experiments, we do not need the phrase transcription to obtain the corresponding alignment because one phrase dependent HMM has been trained with the background partition using a left-to-right model with no skip states, which has been developed with models of 10, 20, 40 and 80 states for each phrase. With these models, we can extract statistics from each utterance of the database and use this alignment information inside our DNN architecture. As inputs to train the alignment models and to the DNN, we employ 20-dimensional MFCC with their first and second derivatives as features to obtain a final input dimension of 60. Then, an energy based voice activity detector is used over them. On these input features, we apply a data augmentation method called Random Erasing [23], which helps us to avoid overfitting in our models due to the scarceness of data in this database.

We have evaluated different configurations of the DNN architecture, which we refer to as the front-end part of the system. In addition, we also have evaluated the interaction of these architectures with the alignment and the utterance duration by means of several HMM sizes. Finally, we have extracted supervectors as a combination of front-end and alignment with a flattened layer, and, with them, we have obtained speaker verification scores by using a cosine similarity metric without any normalization technique.

A set of experiments was performed using Pytorch [24] to evaluate our system. We compare a front-end that uses an average pooling similar to [4,5], and the acoustic feature input directly aligned with the HMM alignment; thus, we obtain the traditional supervector. Additionally, we have evaluated several front-end architectures with four different configurations: one convolutional layer with a kernel of dimension 1 equivalent to a dense layer but keeping the temporal structure and without adding context information, one convolutional layer with a kernel of dimension 3, three convolutional layers with a kernel of dimension 3, and four convolutional layers with a kernel of dimension 3. These configurations for the kernel size and the number of layers have been selected as simple as possible since, as we mentioned in Section 2.2, deep neural networks have not achieved good results in this task.

In addition, in this work, we have used as a reference system the traditional supervector instead of an i-vector based system due to the limitations of i-vectors for text-dependent tasks in the RSR2015 database, as pointed out in [25,26], where the authors concluded that these kinds of datasets do not have enough data to train the traditional i-vectors extractors properly.

## 4. Results

In this work, two sets of experiments have been developed. First, we focus on the RSR-Part I that is composed of fixed pass-phrases, e.g., *"Only lawyers love millionaires"*. In these experiments, we study

the behaviour of our system when we vary the number of front-end layers, the training data and the states of the HMM. The second part of experiments evaluate the performance of the proposed system on the RSR-Part II that is based on short commands with strong overlap lexical content of different commands, e.g., *"Volume up"* and *"Volume down"*.

### 4.1. Experiments with RSR-Part I

In Table 1, we show Equal Error Rate (EER), and National Institute of Standards and Technology (NIST) 2010 minimum detection costs (*DCF10* [27]) results with the different architectures trained on the background subset for female, male and both partitions together. The *DCF10* measures the cost of detection errors in terms of a weighted sum of false alarm and miss probabilities for some decision threshold, and a priori probability of observing target and non-target speakers. We have found that, as we expected, the first approach with an average reduction mechanism for extracting embeddings does not perform well for this text-dependent speaker verification task. It seems that these types of embeddings do not represent correctly the information to achieve discrimination between the correct speaker and phrase both simultaneously. Furthermore, we show that changing the typical average reduction for a new alignment layer inside the DNN achieves a relative improvement of 92.14% in terms of EER%. In addition, we have evaluated the performance of our system in terms of whether we vary the number of states that are used to model each phrase. Given the length of the phrases in Part I, we observe that it is better to employ a greater number of states to correctly model each phrase.

**Table 1.** Experimental results on RSR2015-Part I [22] eval set, where Equal Error Rate (EER%) and National Institute of Standards and Technology (NIST) 2010 min cost (*DCF10*) are shown. These results were obtained by training only with a bkg subset and by varying the number of states of the Hidden Markov Model (HMM). The best results are marked in bold.

| Architecture | | | | Results (EER%/*DCF10*) | | |
|---|---|---|---|---|---|---|
| **Front-End Layers** | **Kernel** | **Pooling Type** | **States** | **Fem** | **Male** | **Fem + Male** |
| 3C | 3 | *avg* | − | 11.20/0.9808 | 12.13/0.9908 | 11.70/0.9887 |
| − | − | *HMM* | 10 | 3.14/0.2733 | 7.18/0.3746 | 6.37/0.4330 |
| 1C | 1 | | | 1.75/0.2759 | 2.58/0.3045 | 2.66/0.3465 |
| 1C | 3 | | | 1.60/0.2504 | 2.31/0.2926 | 2.29/0.3087 |
| 3C | 3 | | | 1.72/0.3150 | 2.78/0.4049 | 2.45/0.3775 |
| 4C | 3 | | | 2.31/0.3840 | 3.28/0.5024 | 3.00/0.4596 |
| − | − | *HMM* | 20 | 1.81/0.2186 | 3.00/0.2197 | 3.20/0.3338 |
| 1C | 1 | | | 1.42/0.1923 | 1.37/0.1904 | 1.88/0.2954 |
| 1C | 3 | | | 1.17/0.1777 | 1.07/0.1722 | 1.51/0.2547 |
| 3C | 3 | | | 1.14/0.1806 | 1.42/0.2715 | 1.34/0.2356 |
| 4C | 3 | | | 1.03/0.1940 | 1.48/0.2712 | 1.34/0.2350 |
| − | − | *HMM* | 40 | 1.35/0.2099 | 1.57/0.1643 | 2.03/0.2886 |
| 1C | 1 | | | 1.16/0.1749 | 0.98/0.1495 | 1.56/0.2774 |
| 1C | 3 | | | 1.04/0.1696 | **0.77/0.1383** | 1.20/0.2335 |
| 3C | 3 | | | 0.86/0.1566 | 1.01/0.2094 | 0.98/0.1940 |
| 4C | 3 | | | 1.09/0.1975 | 1.27/0.2530 | 1.23/0.2402 |
| − | − | *HMM* | 80 | 1.70/0.2352 | 1.73/0.1814 | 2.25/0.3108 |
| 1C | 1 | | | 1.46/0.1971 | 1.19/0.1566 | 1.67/0.2834 |
| 1C | 3 | | | 1.27/0.1887 | 0.88/0.1479 | 1.43/0.2595 |
| 3C | 3 | | | **0.84/0.1533** | 0.88/0.1868 | **0.92/0.1872** |
| 4C | 3 | | | 1.04/0.2096 | 1.18/0.2708 | 1.17/0.2492 |

Nevertheless, the EER results were still quite high, so we decided that the results could be improved by training with background and developing subsets together. In Table 2, we can see that, if we use more data to train our systems, we achieve better performance especially in deep architectures

with more than one layer. This improvement is observed for both architectures. This fact remarks on the importance of having a large amount of data to be able to train deep architectures. In addition, we carried out an experiment to illustrate this effect. In Figure 4, we show that, if we increase little by little the amount of data used to train, the results progressively improve. In addition to that, we can see in Figure 4b that the alignment mechanism makes the system more robust to training data size. To make this representation, we have used the alignment mechanism with 40 states. In addition, the shaded areas of Figure 4 depict the standard deviation of the EER values obtained from each system trained three times.

**Table 2.** Experimental results on RSR2015-Part I [22] eval set, showing EER% and NIST 2010 min cost (*DCF10*). These results were obtained by training with bkg+dev subsets and by varying the number of states of the HMM. The best results are marked in bold.

| Architecture | | | | Results (EER%/*DCF10*) | | |
|---|---|---|---|---|---|---|
| Front-End | | Pooling | | | | |
| Layers | Kernel | Type | States | Fem | Male | Fem+Male |
| 3C | 3 | *avg* | − | 9.11/0.9585 | 8.66/0.9626 | 8.87/0.9629 |
| − | − | *HMM* | 10 | 3.14/0.2733 | 7.18/0.3746 | 6.37/0.4330 |
| 1C | 1 | | | 1.66/0.2522 | 2.30/0.2654 | 2.49/0.3349 |
| 1C | 3 | | | 1.43/0.2267 | 2.01/0.2561 | 2.13/0.2979 |
| 3C | 3 | | | 1.29/0.2342 | 2.08/0.2870 | 1.82/0.2746 |
| 4C | 3 | | | 1.56/0.2853 | 2.41/0.3828 | 2.12/0.3445 |
| − | − | *HMM* | 20 | 1.81/0.2186 | 3.00/0.2197 | 3.20/0.3338 |
| 1C | 1 | | | 1.30/0.1806 | 1.29/0.1625 | 1.79/0.2787 |
| 1C | 3 | | | 1.15/0.1675 | 1.05/0.1397 | 1.51/0.2392 |
| 3C | 3 | | | 0.83/0.1329 | 1.11/0.1897 | 1.03/0.1795 |
| 4C | 3 | | | 0.99/0.1894 | 1.32/0.2548 | 1.22/0.2375 |
| − | − | *HMM* | 40 | 1.35/0.2099 | 1.57/0.1643 | 2.03/0.2886 |
| 1C | 1 | | | 1.17/0.1822 | 0.98/0.1441 | 1.55/0.2673 |
| 1C | 3 | | | 1.07/0.1521 | 0.78/**0.1175** | 1.24/0.2245 |
| 3C | 3 | | | **0.59**/**0.1046** | 0.71/0.1570 | 0.73/**0.1418** |
| 4C | 3 | | | 0.74/0.1584 | 0.89/0.2153 | 0.86/0.1945 |
| − | − | *HMM* | 80 | 1.70/0.2352 | 1.73/0.1814 | 2.25/0.3108 |
| 1C | 1 | | | 1.32/0.2163 | 1.00/0.1367 | 1.60/0.2969 |
| 1C | 3 | | | 1.17/0.1776 | 0.78/0.1289 | 1.34/0.2521 |
| 3C | 3 | | | 0.65/0.1112 | **0.61**/0.1539 | **0.68**/0.1444 |
| 4C | 3 | | | 0.80/0.1648 | 0.83/0.1980 | 0.84/0.1871 |

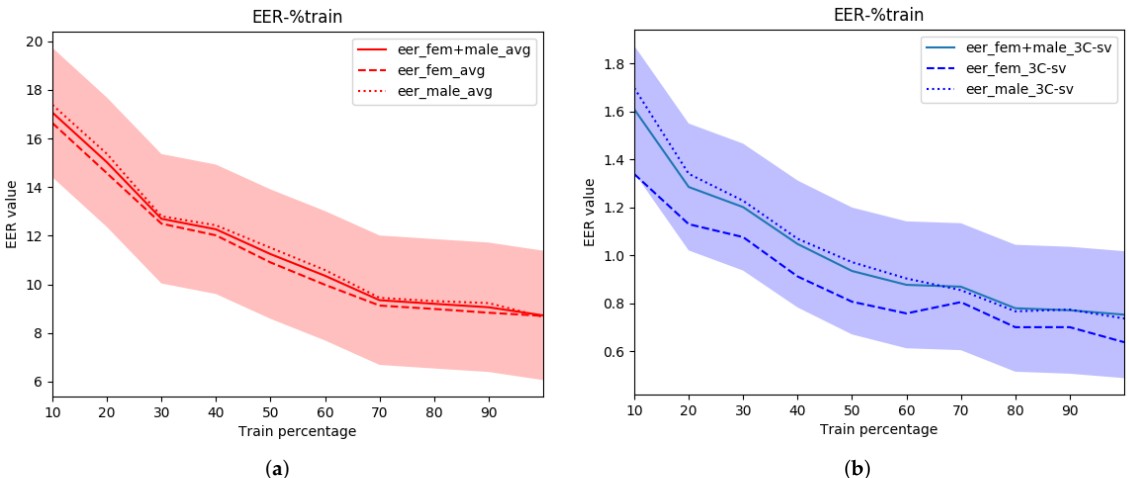

**Figure 4.** Results of EER% varying train percentage where standard deviation is shown only for both gender independent results. (**a**) average embeddings; (**b**) supervectors.

For illustrative purposes, we also represent our high-dimensional supervectors in a two-dimensional space using T-Distributed Stochastic Neighbor Embedding (t-SNE) [28], which preserves distances in a small dimension space. The axes in this representation just define the two-dimensional space to project the high-dimensional space, so they do not have a meaning in terms of units. In Figure 5a, we show this representation for the architecture that uses the time average to extract the embeddings, while, in Figure 5b, we represent the supervectors of our best system. We can see that the second system is able to cluster the examples from the same person, whereas the first method is not able to cluster them together. On the other hand, in both representations, data are auto-organized to show on one side examples from female identities and on the other side examples from male identities.

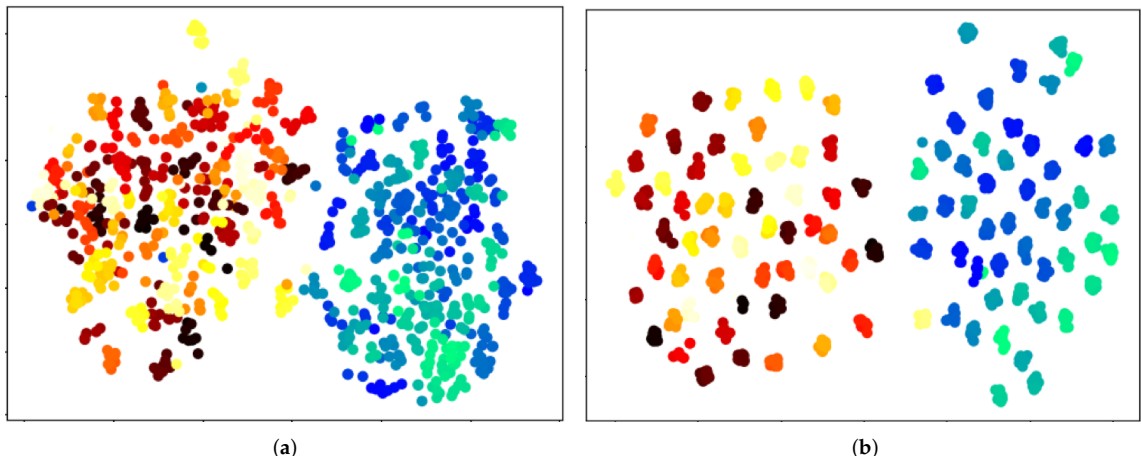

(**a**)               (**b**)

**Figure 5.** Visualizing Average embeddings vs. Supervectors for 1 phrase from male+female using T-Distributed Stochastic Neighbor Embedding (t-SNE), where female is marked by a cold color scale and male is marked by a hot color scale. (**a**) average embeddings; (**b**) supervectors.

Furthermore, we illustrate in Figure 6 the same representation as in Figure 5. However, in this case, we represent the embeddings and the supervectors of the thirty phrases from female identities. With this depiction, we verify something that we had already observed in the previous verification experiments since the embeddings obtained by using a global average pooling layer are not able to separate between the same identity with different phrases and the same identity with the same phrase, which is the base of a text-dependent speaker verification task.

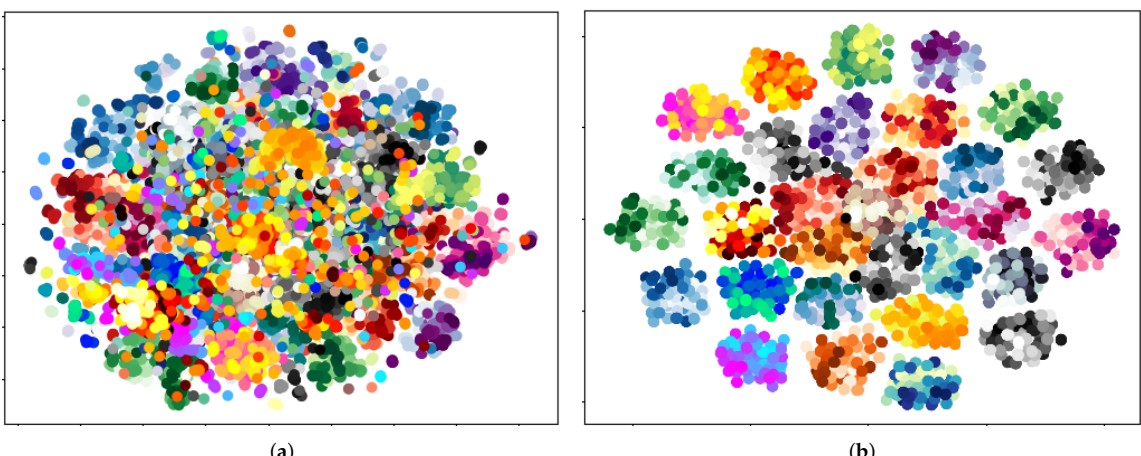

(**a**)               (**b**)

**Figure 6.** Visualizing average embeddings vs. Supervectors for 30 phrases from females using t-SNE. Each phrase is marked by one different color scale. (**a**) average embeddings; (**b**) supervectors.

### 4.2. Experiments with RSR-Part II

The results obtained with Part II are shown in Table 3. In this set of experiments, the phrases are shorter so we can see that we need fewer states to model them. However, the performance obtained with models of 10 and 20 states is similar since some phrases might require more states than others. For example, phrases like *"Turn on coffee machine"* is modelled better with 20 states, while others like *"Call sister"* need only a 10 state model to be characterized.

**Table 3.** Experimental results on RSR2015-Part II [22] eval set, showing EER% and NIST 2010 min cost (*DCF10*). These results were obtained by training with bkg+dev subsets and by varying the number of states of the HMM. The best results are marked in bold.

| Architecture | | | | Results (EER%/*DCF10*) | | |
|---|---|---|---|---|---|---|
| **Front-End** | | **Pooling** | | | | |
| **Layers** | **Kernel** | **Type** | **States** | **Fem** | **Male** | **Fem+Male** |
| 3C | 3 | *avg* | − | 11.18/0.9824 | 11.81/0.9811 | 11.59/0.9813 |
| − | − | *HMM* | 10 | 7.99/0.7062 | 9.47/0.6741 | 9.79/0.7810 |
| 1C | 1 | | | 4.76/0.5901 | 4.70/0.5559 | 5.47/0.6488 |
| 1C | 3 | | | 4.16/0.5466 | **4.31/0.5387** | 4.94/0.6140 |
| 3C | 3 | | | 3.91/**0.5398** | 4.97/0.6536 | 4.68/0.6151 |
| 4C | 3 | | | 4.82/0.6421 | 6.13/0.7092 | 5.73/0.6876 |
| − | − | *HMM* | 20 | 10.06/0.7449 | 10.01/0.7311 | 11.06/0.8032 |
| 1C | 1 | | | 6.44/0.6709 | 5.56/0.6012 | 7.02/0.7403 |
| 1C | 3 | | | 5.57/0.6178 | 5.01/0.5547 | 6.17/0.6865 |
| 3C | 3 | | | **3.90**/0.5420 | 4.86/0.6451 | **4.61/0.6067** |
| 4C | 3 | | | 5.00/0.6539 | 6.23/0.7213 | 5.88/0.6960 |
| − | − | *HMM* | 40 | 18.13/0.8197 | 18.16/0.8146 | 18.43/0.8563 |
| 1C | 1 | | | 11.60/0.7995 | 10.53/0.7493 | 11.74/0.8469 |
| 1C | 3 | | | 9.52/0.7689 | 8.83/0.7039 | 9.95/0.8190 |
| 3C | 3 | | | 5.55/0.6461 | 6.76/0.6830 | 6.59/0.6794 |
| 4C | 3 | | | 7.56/0.7607 | 9.52/0.8147 | 8.93/0.7943 |

Furthermore, as we expected, the general performance is worse since the system suffers from the lexical similarity of the short commands. Thus, this part is more challenging than Part I as we can also see in other previous works [15,16].

In addition to the previous table, Figure 7 depicts the Detection Error Trade-off (DET) curves. These representations show the performance for female+male experiments with the best systems for each pooling configuration. We can observe that the results with the HMMs of 10 or 20 states are practically the same.

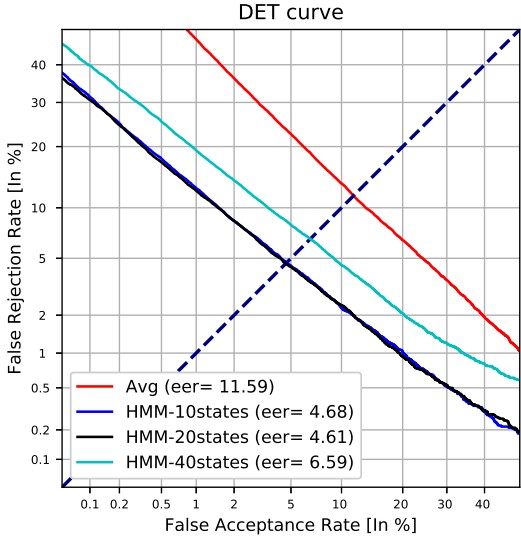

**Figure 7.** Detection Error Trade-off (DET) curve for female+male results on RSR2015-Part II of the best systems for each pooling configuration.

## 5. Conclusions

In this paper, we present a new method to add an alignment layer inside the DNN architectures for encoding meaningful information from each utterance in a supervector, which allows us to conserve the relevant information that we use to verify the speaker identity and the correspondence with the correct phrase. We have evaluated the models in the text-dependent speaker verification database RSR2015-Part I and Part II. Results confirm that the alignment as a layer within the architecture of DNN is an interesting approach since we have obtained competitive results with a straightforward and simple alignment technique that has low computational cost, so we strongly believe that can achieve better results with more powerful techniques such as GMM or DNN posteriors. As a first approximation to the proposed future work, we have an ongoing development using GMM as a new alignment technique producing good preliminary results [29].

**Author Contributions:** Conceptualization, V.M. and A.M.; Investigation, V.M. and A.M.; Methodology, V.M. and A.M.; Software, V.M. and A.M.; Supervision, A.M., A.O. and E.L.; Writing—original draft, V.M.; Writing—review and editing, A.M., A.O. and E.L.

**Funding:** This work has been supported by the Spanish Ministry of Economy and Competitiveness and the European Social Fund through the project TIN2017-85854-C4-1-R, by the Government of Aragon (Reference Group T36_17R) and co-financed with Feder 2014-2020 "Building Europe from Aragon", and by Nuance Communications, Inc. (Burlington, MA, USA).

**Acknowledgments:** We gratefully acknowledge the support of NVIDIA Corporation (Santa Clara, CA, USA) with the donation of the Titan Xp GPU used for this research.

**Conflicts of Interest:** The authors declare no conflict of interest.

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
