# Peer review of "Supervector Extraction for Encoding Speaker and Phrase Information with Neural Networks for Text-Dependent Speaker Verification†"

_applsci, doi:10.3390/app9163295_

Round 1
Reviewer 1 Report
1. In this paper, the authors propose a supervector extraction algorithm using 1-dimension CNN and HMM. The proposed supervector extraction algorithm conserve the relevant information that use to verify the speaker identity and the correspondence with the correct phrase. The results of figure 5 and 6 are show that good performance of the data segmentation.
2. The authors use of 1-dimension CNN to extract the features and operate in the temporal dimension to add context information to the process and at the same time the channels are combined at each layer. The context information which is added depends on the size of the kernel used in the convolution layer. On the other hand, in addition to the size of the kernel, the number of the convolutional layers is also used to compare the experimental results. Although the experimental results show the good performance, but how to decide the parameters, kernel size and layer number, becomes a problem. It is recommended that the author propose a feasible setting method.
3. The experimental results compare to the average pooling of the proposed supervector extraction, but there is a lack of comparison of traditional feature extraction methods. It is recommended to compare the traditional feature extraction methods to prove the advantages of this proposed method.

Author Response
We would like to thank the reviewers for their efforts and comments which we think have served to improve the quality of the paper. We have updated the manuscript to reflect the modifications motivated by these observations.
Comments of Reviewer 1
Point 1: The authors use of 1-dimension CNN to extract the features and operate in the temporal dimension to add context information to the process and at the same time the channels are combined at each layer. The context information which is added depends on the size of the kernel used in the convolution layer. On the other hand, in addition to the size of the kernel, the number of the convolutional layers is also used to compare the experimental results. Although the experimental results show the good performance, but how to decide the parameters, kernel size and layer number, becomes a problem. It is recommended that the author propose a feasible setting method.
Response 1: To address this comment, we have included the next sentence in the last paragraph in Section 3.2.
These configurations for the kernel size and the number of layers have been selected as simple as possible since as we mentioned in Section 2.2., deep neural networks have not achieved good results in this task.
Point 2: The experimental results compare to the average pooling of the proposed supervector extraction, but there is a lack of comparison of traditional feature extraction methods. It is recommended to compare the traditional feature extraction methods to prove the advantages of this proposed method.
Response 2: We agree with reviewer 1 that it is interesting to compare our proposed method with other traditional feature extraction methods. Thus, in the experiments we added an experiment based on the traditional supervector method which consists on aligned directly the acoustic feature with the HMM alignment, so we did not use any neural network for that process. On the other hand, we considered adding a traditional i-vector extractor as the baseline system, but in [25] and [26] is explained that the traditional i-vector extractors usually do not perform well in text-dependent speaker recognition because databases such as RSR2015 have insufficient training data to train i-vector extractors on text constrained tasks with more than just one passphrase.
We have added a new paragraph at the end of Section 3.2 to clarify our choice about the reference system without neural networks employed.
Besides, in this work, we have used as reference system the traditional supervector instead of an i-vector based system due to the limitations of i-vectors for text-dependent tasks in the RSR2015 database, as pointed out in [25,26] where the authors concluded that these kind of datasets have not enough data to train the traditional i-vectors extractors properly.

Reviewer 2 Report
The authors present a system for text-dependent speaker verification based upon a Deep Neural Networks and an HMM for alignment.
The english language are fine but my suggestion is to clarify some aspects of the problem for readers that are not expert of the field and are interested in the method.
I suggest to introduce the following terms in the introduction: embeddings, supervector, i-vector
I suggest to introduce the DCF10 metric in the experimental Results. EER is a general and well know metric, DCF10 is not.
How many states are used for plotting Figure 4.(b)?
Please, report x and y labels in Figures 5 and 6
Even if Figures 5 and 6 are very impressive some numerical values are required. Please, reports the number of samples that are correctly and wrongly associated to the right cluster.
A comparison with the state of the art is useful. Please, report the performance of the top performing systems on the same datasets.
Part of the future works reported in the Conclusion are presented in a paper of the same authors available in arxiv https://arxiv.org/pdf/1901.11332.pdf Cite this paper.
Author Response
We would like to thank the reviewers for their efforts and comments which we think have served to improve the quality of the paper. We have updated the manuscript to reflect the modifications motivated by these observations.
Comments of Reviewer 2
Point 1: I suggest to introduce the following terms in the introduction: embeddings, supervector, i-vector.
Response 1: We agree with reviewer 2 that some terms need extra explanation for readers that are not expert in the field. For that reason, the next sentences in the Introduction section have been added or modified.
In paragraph 1:
As we will show, it is a more natural solution for the text-dependent speaker verification since the speaker and phrase information can be encoded in the supervector thanks to the neural network and the specific states of the supervector. A supervector is a concatenation of smaller-dimensional vectors of each specific state into a higher-dimensional vector.
In paragraph 2:
In the context of text-independent speaker verification tasks, the baseline system based on i-vector extraction and Probabilistic Linear Discriminant Analysis (PLDA) [6,7] are still among the best results of the state-of-the-art. An i-vector is a representation of an utterance in a low-dimensional subspace called the total variability subspace, and the PLDA model produces the verification scores. However, as we previously…
Once these systems are trained, the embeddings, which are fixed-length utterance level representations, are extracted by reduction mechanisms from the DNN [11-13]. In this context, they are called x-vectors.
Point 2: I suggest to introduce the DCF10 metric in the experimental Results. EER is a general and well know metric, DCF10 is not.
Response 2: We have added a brief definition of DCF10 metric to explain it in the first paragraph in Section 4.1.
In Table 1, we show equal error rate (EER), and the NIST 2010 minimum detection costs (DCF101) results with the different architectures trained on the background subset for female, male and both partitions together. The DCF10 measures the cost of detection errors in terms of a weighted sum of false alarm and miss probabilities for some decision threshold, and a priori probability of observing target and non-target speakers.
Point 3: How many states are used for plotting Figure 4.(b)?
Response 3: We have modified paragraph 2 in Section 4.1 to add the number of states used in Figure 4.(b).
In addition to that, we can see in Fig.4(b) that the alignment mechanism makes the system more robust to training data size. To make this representation, we have used the alignment mechanism with 40 states. Besides, the shaded areas of Fig.4 depict the standard deviation of the EER values obtained from each system trained three times.
Point 4: Please, report x and y labels in Figures 5 and 6
Response 4: We have not reported labels for the x and y axes because the axes in t-SNE representations have no interpretable meaning in terms of units. The axes just define a 2D space to project the higher dimensional space, while the distances (according to some metric) between the high-dimensional data are intended to be preserved in the low-dimensional space. To clarify this representation, we have removed the values in the axes in Figures 5 and 6, as is usually made in other works when they make this kind of representations as in:
Besides, we have added the next sentence in paragraph 3 in Section 4.1 to remark this information about t-SNE representations:
For illustrative purposes, we also represent our high-dimensional supervectors in a two-dimensional space using t-SNE [27] which preserves distances in a small dimension space. The axes in this representation just define the two-dimensional space to project the high-dimensional space, so they do not have a meaning in terms of units.
Point 5: Even if Figures 5 and 6 are very impressive some numerical values are required. Please, reports the number of samples that are correctly and wrongly associated to the right cluster.
Response 5: We have only employed Figures 5 and 6 as interpretable representations of our results reported in Tables 1, 2 and 3 using speaker verification metrics. These representations are subjective but more understandable for readers that are not expert in the field and the metrics usually employed in speaker verification.
Point 6: A comparison with the state of the art is useful. Please, report the performance of the top performing systems on the same datasets.
Response 6: We agree with reviewer 2, but the experimental setup usually differ and most of the systems previously presented use external data to train their systems. To present a fair comparison, in our case we compare to baseline systems that use the same amount of training and we compare whether using or not the presented technologies presents improvements.
Point 7: Part of the future works reported in the Conclusion are presented in a paper of the same authors available in arxiv
https://arxiv.org/pdf/1901.11332.pdf Cite this paper.
Response 7: We have added a new sentence in the Conclusion section to reference our other work, which has been overlapped in time due to this work is an extended version of a previous conference paper.
Results confirm that the alignment as a layer within the architecture of DNN is an interesting approach since we have obtained competitive results with a straightforward and simple technique which has low computational cost, so we strongly believe that can achieve better results with more powerful techniques such as Gaussian Mixture Model (GMM) or DNN posteriors. As a first approximation to the proposed future work, we have an ongoing development using GMM as new alignment technique producing good preliminary results [28].
